

# Patterns in artisanal coral reef fisheries revealed through local monitoring efforts

David G. Delaney[1,2], Lida T. Teneva[3], Kostantinos A. Stamoulis[1,4], Jonatha L. Giddens[1], Haruko Koike[1,5], Tom Ogawa[5], Alan M. Friedlander[1,6] and John N. Kittinger[3,7]

[1] Department of Biology, Fisheries Ecology Research Laboratory, University of Hawai'i, Honolulu, HI, USA
[2] Delaney Aquatic Consulting LLC, Honolulu, HI, USA
[3] Center for Oceans, Conservation International, Honolulu, HI, USA
[4] Department of Environment and Agriculture, Curtin University, Perth, WA, Australia
[5] Division of Aquatic Resources (DAR), Department of Land and Natural Resources, Honolulu, HI, USA
[6] Pristine Seas, National Geographic Society, Washington, DC, USA
[7] Center for Biodiversity Outcomes, Julie Ann Wrigley Global Institute of Sustainability, Arizona State University, Tempe, AZ, USA

Corresponding author
David G. Delaney,
dgdelaney@gmail.com

## ABSTRACT

Sustainable fisheries management is key to restoring and maintaining ecological function and benefits to people, but it requires accurate information about patterns of resource use, particularly fishing pressure. In most coral reef fisheries and other data-poor contexts, obtaining such information is challenging and remains an impediment to effective management. We developed the most comprehensive regional view of shore-based fishing effort and catch published to date, to show detailed fishing patterns from across the main Hawaiian Islands (MHI). We reveal these regional patterns through fisher "creel" surveys conducted by local communities, state agencies, academics, and/or environmental organizations, at 18 sites, comprising >10,000 h of monitoring across a range of habitats and human influences throughout the MHI. All creel surveys included in this study except for one were previously published in some form (peer-reviewed articles or gray literature reports). Here, we synthesize these studies to document spatial patterns in nearshore fisheries catch, effort, catch rates (i.e., catch-per-unit-effort (CPUE)), and catch disposition (i.e., use of fish after catch is landed). This effort provides for a description of general regional patterns based on these location-specific studies. Line fishing was by far the dominant gear type employed. The most efficient gear (i.e., highest CPUE) was spear (0.64 kg h$^{-1}$), followed closely by net (0.61 kg h$^{-1}$), with CPUE for line (0.16 kg h$^{-1}$) substantially lower than the other two methods. Creel surveys also documented illegal fishing activity across the studied locations, although these activities were not consistent across sites. Overall, most of the catch was not sold, but rather retained for home consumption or given away to extended family, which suggests that cultural practices and food security may be stronger drivers of fishing effort than commercial exploitation for coral reef fisheries in Hawai'i. Increased monitoring of spatial patterns in nearshore fisheries can inform targeted management, and can help communities develop a more informed understanding of the drivers of

marine resource harvest and the state of the resources, in order to maintain these fisheries for food security, cultural practices, and ecological value.

## INTRODUCTION

Fisheries contribute 20% of the protein for >3 billion people and 17% of global protein consumed, representing a crucial contribution to global food security (*UN FAO, 2016*). In the tropics, coral reef fisheries support >6 million reef fishers in over 100 countries, providing critical and diverse services, including food, income, livelihoods, and cultural significance (*Teh, Teh & Sumaila, 2013*). Nowhere are coral reef fisheries more important than in the developing economies and communities in the Pacific (*Dalzell, 1996*; *Gillett, 2016*). In Hawai'i, these fisheries are relied upon for economic, social, and cultural services, including important livelihood and food provisioning (*Friedlander, Shackeroff & Kittinger, 2013*). Approximately a third of Hawai'i residents identify themselves as fishers, and the diversity of cultures that live in Hawai'i all place a high importance on fishing (*OmniTrak, 2011*).

Despite their importance, many small-scale reef fisheries, both commercial and non-commercial, in the Pacific have significant capacity gaps in management, threatening the food security and livelihoods that these fisheries provide to communities (*Newton et al., 2007*; *Bell et al., 2009*; *Kronen et al., 2010a, 2010b*; *Houk et al., 2012*; *Friedlander, Nowlis & Koike, 2014*). Many of the challenges currently hindering sustainable management and fisheries sector development strategies are associated with a lack of information for these multi-species, multi-gear small-scale coral reef fisheries (*Cinner et al., 2012*; *Fenner, 2012*).

One of the most persistent knowledge gaps for scientists and managers surrounds the dominant harvesting modes and magnitude of current fishing activities, including the total production of the fishery and its value to local communities and economies. This gap is largely due to a lack of assessments of fishing activities and fish stocks, a challenge that is common in un-assessed or otherwise data-poor fisheries, which account for more than 80% of the global fisheries catch (*Dalzell, 1996*; *Sale, 2008*; *Costello, Wilson & Houlding, 2012*; *Ricard et al., 2012*; *Friedlander, 2015*). The lack of investment in monitoring is primarily due to low technical and financial capacity in many coral reef geographies, as well as the complexity of these small-scale fisheries, which precludes an accurate understanding of fisheries status, which is required to develop effective, evidence-based regulations (*Pauly, 2006*; *Zeller et al., 2006*; *Pauly & Zeller, 2014*).

To develop better management strategies, scientists and managers need more accurate estimates of how much fish biomass is in the water, how much is being fished, what fishing gears are used, and whether the rates and amount of catch are ecologically sustainable. A variety of methods have been advanced to address this gap (*Nadon et al., 2015*;

*Prince et al., 2015*; *Anderson et al., 2017*; *Nadon, 2017*; *Rosenberg et al., 2017*), but the data necessary to accomplish these assessments remain a major limitation. An empirical method for fisheries assessment that has worked effectively at local community scales is the creel survey approach, which focuses on estimating total catch, gear types used, selectivity of gear types (i.e., variety in targeted species), and other aspects of fishing behavior (*Malvestuto, 1983*). Creel surveys, utilizing estimates of fishing effort paired with fisher interviews, have been particularly useful in assessing the nearshore fisheries and total economic value to local communities in several locations in the Pacific (*Albert et al., 2015a*, *2015b*; *Weijerman et al., 2016*), and particularly in Hawai'i (*Friedlander & Parrish, 1997*; *Everson & Friedlander, 2004*; *Kittinger et al., 2015*). The name 'creel survey' comes from the woven basket, or creel, that freshwater anglers use to hold their catch (*Malvestuto, 1996*). In Hawai'i, these surveys are referred to by the Hawaiian name for a tin basin used to hold nets called a pakini (*Kittinger et al., 2015*). Surveys are typically conducted at access points where fishers are asked about their fishing activities. This approach is generally more effective than studies that do not engage the local community, because the information gathered stems from the fishers themselves (*Whyte, Greenwood & Lazes, 1989*; *Scholz et al., 2004*; *Kittinger, 2013*). Some creel surveys, especially for more recent survey efforts, have been led by communities, in a "co-design" format. Creel surveys offer tremendous advantages in terms of accuracy at a particular location. However, they are resource-intensive to implement, and are therefore often limited in their spatial and temporal scope, precluding researchers, managers, and communities from recognizing larger-scale trends necessary for managing targeted fish stocks in reef fisheries (*Weijerman et al., 2016*).

The purpose of our study was to gather, collate, and synthesize with unparalleled geographical coverage and detail, a clearer picture of the reef-associated fishing effort, catch, catch-per-unit-effort (CPUE), and fate of the reef catch at a regional-scale, using a unique dataset and case study approach in Hawai'i. This work addresses long-standing interest in information about coastal fisheries in Hawai'i and similar coral reef geographies, which have remained poorly quantified, particularly in terms of non-commercial fishing effort and catch. The patterns found here are determined through, arguably, the most accurate and high-resolution methodology: through fisher surveys conducted at 18 sites across the archipelago. This broad regional coverage provides unique insights into the current state of nearshore fishing effort and catch, and demonstrates the value of creel surveys as a community-level monitoring technique producing information critical to effective fisheries management by assessing distinct spatial patterns in: (1) gear usage; (2) annual catch; and (3) disposition of the catch.

## METHODS

### Study sites

Hawai'i has a population of approximately 1.4 million people, with ~70% of the population residing on the island of O'ahu (Fig. 1). The remaining population varies widely across the other islands, with 144,444 people on Maui, 185,079 on Hawai'i Island, 66,921 on Kaua'i, and only 7,345 people on Moloka'i (*State of Hawai'i, 2010*).

Wave energy varies spatially and seasonally (*Gove et al., 2013*), and has a strong influence on the composition of the nearshore biological communities (*Dollar, 1982*; *Friedlander et al., 2003*), as well as on fishing effort, with north facing shores having reduced fishing during the winter months when wave activity is at its highest (*Stamoulis & Friedlander, 2013*).

Current monitoring of nearshore fisheries by state and federal agencies in the main Hawaiian Islands (MHI) does not capture the full extent of fishing catch and effort due to the non-commercial nature of these fisheries, which unlike commercial catch is not required to be reported, and the disparate landing sites across the state (*Smith, 1993*; K. McCoy et al., 2015, unpublished data). To address this issue, we complied a large database of creel surveys conducted at 18 sites, across several islands, and over three decades. Details on the compiled information from creel surveys for fishing effort, catch, CPUE, fish flows (i.e., catch disposition), and illegal fishing (e.g., use of illegal gear, take of undersized regulated species, fishing in restricted areas) for the MHI and reported values are presented in Supplemental Information S1 and references in Table 1. All creel surveys included in this study were previously published in some form (peer-reviewed articles or gray literature reports) except the data for Kaloko-Honokōhau National Historical Park, Hawai'i Island, which came from intercept interview data (i.e., interview data with fishers), from which we produced estimates of effort, catch, and CPUE that are described in Supplemental Information S1.

Study sites were included in this study based on the following four criteria. First, a creel and/or fish flow survey was conducted at the site in the MHI. Second, we had access to a report, publication, and/or the raw data. Third, for creel surveys, the monitoring effort had to be conducted for longer than one month. Fourth, that the study met data quality standards based on prior knowledge by the authors in consultation with other local authorities. Sample sites included urban and major tourist destinations, such as Maunalua Bay, Pearl Harbor, and Waikīkī on O'ahu, as well as Wailuku on Maui, which we expected to be characterized by high level of fishing effort but low catch based on anecdotal evidence. Conversely, remote rural communities such as Kalaupapa National Historical Park on Moloka'i, and Hā'ena on Kaua'i, were expected to have higher catch rates but lower overall effort (Fig. 1). Puakō on Hawai'i Island was surveyed from May 1980 to September 1981, and again from December 2008 to December 2009 (Table 1).
For patterns in fishing gear, we used data from all 18 locations. Examination of fishing effort and total catch had 14 available datasets, with 13 for CPUE, and 8 for fish flow (Table 1).

## Creel survey methodology

Creel surveys in Hawai'i have typically quantified fishing effort using interviews and/or elevated vantage points, where observers scanned the area on a systematic schedule using binoculars and/or high-power spotting telescopes (*Friedlander & Parrish, 1997*; *Tom, 2011*; *Friedlander et al., 2014a*). Interview-based surveys are conducted using access point and roving survey methods. An access point survey targets a specific site that generally has a single pathway where fishers can be sampled upon completion of a fishing trip

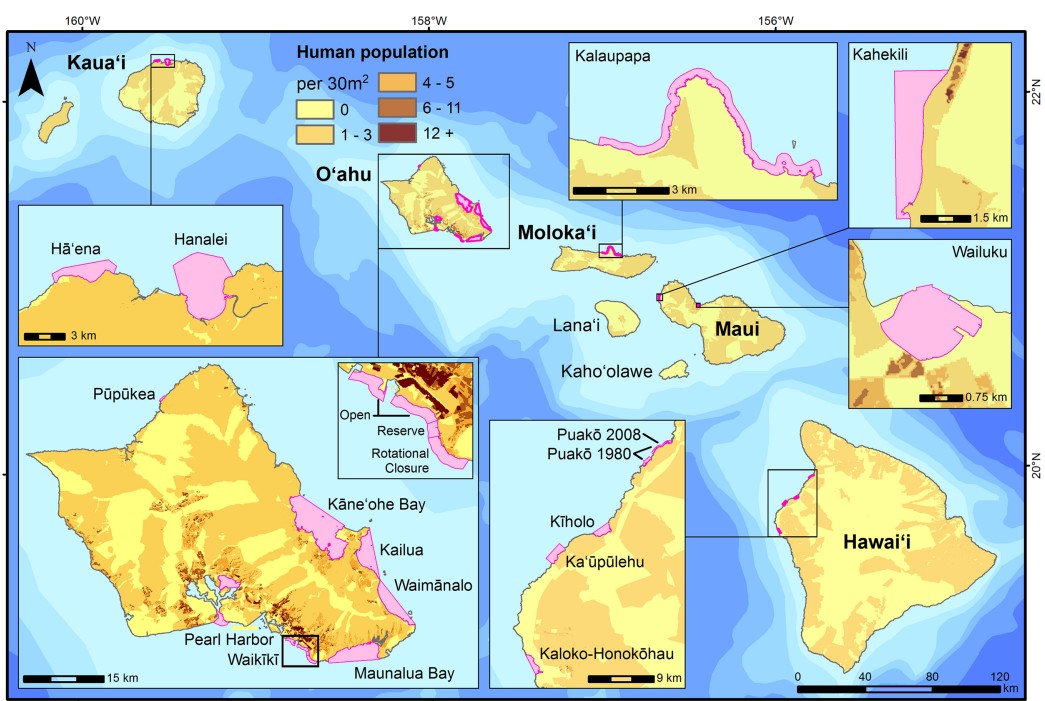

**Figure 1 Survey sites where creel and/or fish flow surveys were conducted and included in this study are shown in pink.** 2010 human population (*State of Hawai'i, 2010*) is distributed based on land cover types within census blocks.

(e.g., piers, jetties, or a remote beach with one entry point (*Robson & Jones, 1989*)). A roving survey targets a broader area where access is generally undefined and fishers are more dispersed (*Malvestuto, Davies & Shelton, 1978*; *Malvestuto, 1996*). It is conducted by walking and/or driving along a stretch of coastline and stopping when a fisher is located for a potential interview (*Malvestuto, Davies & Shelton, 1978*; *Pollock et al., 1997*). Interviews were typically conducted with fishers to gather information such as catch and species composition. Estimates of total annual catch were obtained by multiplying effort for each gear type with the corresponding CPUE and expanded based on a stratified random survey design (references in Table 1 or Supplemental Information S1).

Our assessment was focused on patterns in nearshore fishing by compiling estimates from previously conducted creel surveys to produce regional maps of total annual catch, CPUE, and effort for the three dominant nearshore gear types: shore-based line, net, and spear. These broad categories of gear are the most common and popular types of fishing methods in Hawai'i. We created boundary polygons using ArcGIS 10.4 based on maps and description of surveyed areas in each creel study. These polygons represent the marine area surveyed and were delineated based on the National Oceanic and Atmospheric Administration (NOAA) Biogeography Branch shoreline data for the MHI (*Battista, Costa & Anderson, 2007*). Total area in km$^2$ for each creel survey area polygon was calculated in ArcGIS, as well as area of coral reef and hard bottom as delineated by *Battista, Costa & Anderson (2007)*. Length of shoreline for each creel
**Table 1 Location, availability of data, and its inclusion in analyses of gear most commonly used, effort, catch, CPUE estimates, and/or fish flows and the source for this information.**

| Location | Most popular gear | Effort | Catch | CPUE | Fish flow | Source |
|---|---|---|---|---|---|---|
| Hāʻena, Kauaʻi | 1 | 0 | 0 | 1 | 1 | *Vaughan & Vitousek (2013)*, Kuaʻāina Ulu ʻAuamo (KUA), Hui Makaʻāinana o Makana and Limahuli Gardens staff, 2010, unpublished report, and K. McCoy et al., 2015, unpublished data |
| Hanalei, Kauaʻi | 1 | 1 | 1 | 1 | 1 | *Friedlander & Parrish (1997)*; *Everson & Friedlander (2004)*; *Glazier & Kittinger (2012)* |
| Kahekili, Maui | 1 | 1 | 1 | 1 | 0 | *Friedlander et al. (2012)* |
| Kailua, Oʻahu | 1 | 1 | 0 | 0 | 0 | *Friedlander et al. (2014a)* |
| Kalaupapa, Molokaʻi | 1 | 0 | 0 | 0 | 0 | *Tom (2011)* |
| Kaloko-Honokōhau, Hawaiʻi | 1 | 1 | 1 | 1 | 0 | K. Tom & J. Beets, 2011, unpublished data |
| Kāneʻohe Bay, Oʻahu | 1 | 1 | 1 | 1 | 1 | *Everson & Friedlander (2004)* |
| Kaʻūpūlehu, Hawaiʻi | 1 | 1 | 1 | 1 | 0 | *Koike et al. (2015)* |
| Kīholo, Hawaiʻi | 1 | 1 | 1 | 1 | 1 | *Kittinger et al. (2015)* |
| Maunalua Bay, Oʻahu | 1 | 1 | 1 | 1 | 1 | *Kittinger (2013)* and K. McCoy et al., 2015, unpublished data |
| Pearl Harbor, Oʻahu | 1 | 1 | 1 | 1 | 0 | *Wolfe et al. (2017)* |
| Puakō, Hawaiʻi (1980–1981) | 1 | 1 | 1 | 1 | 0 | *Hayes et al. (1982)* |
| Puakō, Hawaiʻi (2008–2009) | 1 | 1 | 1 | 1 | 1 | *Giddens (2010)* and J. Giddens, 2017, personal communication |
| Pūpūkea, Oʻahu | 1 | 1 | 0 | 0 | 0 | *Stamoulis & Friedlander (2013)* |
| Waikīkī reserve, Oʻahu | 1 | 0 | 1 | 0 | 0 | *Meyer (2003)* |
| Waikīkī open, Oʻahu | 1 | 0 | 1 | 1* | 0 | *Meyer (2003)* |
| Waikīkī rotational closure area, Oʻahu | 1 | 0 | 1 | 0 | 0 | *Meyer (2003)* |
| Wailuku, Maui | 1 | 1 | 1 | 1 | 1 | H. Koike, J. Carpio, A.M. Friedlander, 2014, unpublished data (Final Creel Survey Report for Wailuku Community Management Area, Maui County, Hawaiʻi) and H. Koike, 2017, personal communication |
| Waimānalo, Oʻahu | 1 | 1 | 0 | 0 | 0 | *Friedlander et al. (2014a)* and K. Stamoulis, 2017, personal communication |
| Hawaiʻi Island, Hawaiʻi | 0 | 0 | 0 | 0 | 1 | *Hardt (2011)* |

**Notes:**
"0" indicates the information was not available and "1" indicates the information was available.
* The CPUE estimates for Waikīkī were not reported for the three individual sites separately.

area was measured after first simplifying creel polygon features to standardize measurements using the ArcGIS simplify polygon tool with a maximum allowable offset of 100 m, which removed extraneous bends while preserving the essential shape (Table 2; ArcGIS Desktop: Release 10; Environmental Systems Research Institute, Redlands, WA, USA).

## Fish flow surveys

Improvements in fisheries management are most effective if they are informed by the main drivers of fishing (e.g., commerce, recreation, subsistence, culture). To accomplish this, we obtained information on fish flow across the MHI. This information included the distribution of catch, and whether it was: (1) kept for home

**Table 2 Location, start and end dates of surveys, coastline length, total area, and area of coral reef and hard bottom in creel survey sites as delineated by _Battista, Costa & Anderson (2007)_.**

| Location | Start and end dates | Coastline (km) | Total area (km²) | Area of coral reef and hard bottom (km²) |
|---|---|---|---|---|
| Hāʻena, Kauaʻi | Aug 09–Dec 10 | 3.6 | 2.05 | 1.25 |
| Hanalei, Kauaʻi | Jul 92–Dec 93 | 6.2 | 7.58 | 2.82 |
| Kahekili, Maui | Jan 11–Dec 11 | 3.6 | 1.88 | 0.46 |
| Kailua, Oʻahu | Jan 08–Aug 13 | 11.8 | 14.84 | 12.55 |
| Kalaupapa, Molokaʻi | Aug 08–Nov 10 | 19.6 | 7.41 | 3.08 |
| Kaloko-Honokōhau, Hawaiʻi | Jan 10–Jan 11 | 6.0 | 2.26 | 1.62 |
| Kāneʻohe Bay, Oʻahu | Spring 91–Spring 92 | 33.7 | 48.46 | 23.97 |
| Kaʻūpūlehu, Hawaiʻi | Aug 13–Aug 14 | 3.7 | 3.53 | 2.13 |
| Kīholo, Hawaiʻi | May 12–Apr 13 | 4.5 | 2.65 | 1.77 |
| Maunalua Bay, Oʻahu | Dec 07–Nov 08[*] and Jan 11–Jul 11[#] | 15.1 | 19.12 | 16.11 |
| Pearl Harbor, Oʻahu | Jun 15–May 16 | 14.9 | 8.06 | 1.96 |
| Puakō, Hawaiʻi (1980–1981) | May 80–Sep 81 | 6.6 | 1.43 | 1.27 |
| Puakō, Hawaiʻi (2008–2009) | Dec 08–Dec 09 | 4.9 | 0.85 | 0.75 |
| Pūpūkea, Oʻahu | Jun 11–Sep 11 | 1.2 | 0.31 | 0.30 |
| Waikīkī reserve, Oʻahu | Jun 98–Aug 01 | 0.7 | 0.31 | 0.28 |
| Waikīkī open, Oʻahu | Jun 98–Aug 01 | 4.8 | 1.80 | 1.41 |
| Waikīkī rotational closure area, Oʻahu | Jun 98–Aug 01 | 1.9 | 0.97 | 0.84 |
| Wailuku, Maui | Mar 13–May 14 | 3.3 | 0.93 | 0.16 |
| Waimānalo, Oʻahu | Jan 08–Aug 13 | 11.4 | 14.22 | 6.15 |

Notes:
Surveys were conducted from 1980 to 2016.
[*] Start and end dates for the creel survey.
[#] Start and end dates for the fish flow survey.

consumption; (2) given away; (3) sold (or bartered); (4) released; (5) used as bait and/or (6); used for other purposes (_Hardt, 2011_; _Kittinger, 2013_; _Kittinger et al., 2015_). Fish flow information estimates how catch from nearshore marine ecosystems is used by local fishers and the role it has in local economies and households (_Glazier & Kittinger, 2012_; _Kittinger, 2013_).

# RESULTS

## Patterns in effort, catch and CPUE from creel surveys

Line fishing was the most commonly employed gear type at all the sites except for one (94% of the sites; Table 3), with net fishing (primarily cast nets) being most commonly used at Hāʻena, Kauaʻi (surveyed 2009–2010) (6% of the sites; Table 3). In all cases where the estimate of fishing effort was quantified in hours, line fishing had the highest estimate of effort (Table 3). On average, line fishing was almost 80% of the total shore fishing effort with only 7% and 14% from net and spear fishing, respectively. However, the most efficient gear types (i.e., highest CPUE) were spear ($\bar{X} = 0.64$ kg h$^{-1}$; SE = 0.12), followed closely by net ($\bar{X} = 0.61$ kg h$^{-1}$; SE = 0.19), with CPUE for line

**Table 3 Location, most commonly used fishing gear type ("gear:" gear with highest frequency of occurrence or density of fishing activities by gear type), estimates of effort for three shore-based fishing gear types (h), total annual catch (kg), percent of total catch that is biomass of *Selar crumenophthalmus* ("scad"), and octopus (*Octopus cyanea* and *Callistoctopus ornatus*).**

| Location | Gear | Line | Net | Spear | Catch | % Scad | % Octopus |
|---|---|---|---|---|---|---|---|
| Hā'ena, Kaua'i | Net | – | – | – | – | – | – |
| Hanalei, Kaua'i | Line | 15,850 | 5,370 | 397 | 15,801 | 39.4 | – |
| Kahekili, Maui | Line | 3,925 | 108 | 2,857 | 1,214 | – | 36.6 |
| Kailua, O'ahu | Line | 3,867 | 106 | 2,184 | – | – | – |
| Kalaupapa, Moloka'i | Line | – | – | – | – | – | – |
| Kaloko-Honokōhau, Hawai'i | Line | 4,538 | 208 | 2,331 | 3,277 | 0.0 | 5.9 |
| Kāne'ohe Bay, O'ahu | Line | 35,748 | 5,711 | 15,926 | 63,958 | 1.6 | 21.3 |
| Ka'ūpūlehu, Hawai'i | Line | 5,089 | 1,319 | 4,587 | 4,599 | 1.2 | 12.2 |
| Kīholo, Hawai'i | Line | 5,004 | 1,580 | 799 | 7,353 | – | – |
| Maunalua Bay, O'ahu | Line | 16,441 | 888 | 4,099 | 5,543 | – | – |
| Pearl Harbor, O'ahu | Line | 98,725 | 698 | 927 | 7,726 | 0.0 | 2.4 |
| Puakō, Hawai'i (1980–1981) | Line | 5,017 | – | 1,962 | 8,063 | 0.0 | 2.7 |
| Puakō, Hawai'i (2008–2009) | Line | 2,917 | 1,239 | 1,958 | 2,323 | 0.0 | – |
| Pūpūkea, O'ahu | Line | 3,685 | 5 | 1,511 | – | – | – |
| Waikīkī reserve, O'ahu | Line | – | – | – | 28 | – | – |
| Waikīkī open, O'ahu | Line | – | – | – | 457 | – | – |
| Waikīkī rotational closure, O'ahu | Line | – | – | – | 581 | – | – |
| Wailuku, Maui | Line | 15,701 | 2,192 | 719 | 2,161 | 6.3 | 3.1 |
| Waimānalo, O'ahu | Line | 7,140 | 11 | 317 | – | – | – |

**Notes:**
The two species were included since at some sites they compose a large portion of the total harvest. Some values were not available ("–"). Details for derivations of these values are provided in Supplemental Information S1.

($\bar{X} = 0.16$ kg h$^{-1}$; SE = 0.04) 3.9 times lower than spear and 3.7 times lower than net (Table 4).

Fishing effort estimates varied within and across the MHI. The highest estimates of effort were on O'ahu, which is the most populated island in the state (Figs. 1 and 2). The highest estimate of fishing effort was recorded at Pearl Harbor (surveyed 2015–2016), a densely-populated embayment in urban Honolulu, with >100,000 h of non-vessel-based fishing effort, nearly all of which was line fishing (Fig. 2; Table 3). Kāne'ohe Bay (surveyed 1991–1992) is a large, sheltered bay on windward O'ahu and had the second highest total effort among all locations (Fig. 2; Table 3). Effort estimates across O'ahu were extremely variable, with the lowest overall fishing effort observed at Pūpūkea (surveyed 6/2011–9/2011), on the relatively less populated north shore (Figs. 1 and 2). Fishing effort was generally lower at less populated parts of the MHI (Figs. 1 and 2).

Total catch was highly variable among islands (Fig. 3; Table 3), with the highest catch recorded in Kāne'ohe Bay (surveyed 1991–1992) on O'ahu. Catch was generally low in urban and/or tourist-dominated sites such as Waikīkī (surveyed 1998–2001), Pearl Harbor (surveyed 2015–2016), and Maunalua Bay (surveyed 2007–2008) on O'ahu, as well as Kahekili (surveyed 1/2011–12/2011) and Wailuku (surveyed 2013–2014) on Maui. Catches were similar for the four survey sites along the west coast of Hawai'i Island

**Table 4 Catch-per-unit-effort (CPUE) estimates in kg h$^{-1}$ for three shore-based fishing gear types (line, net, and spear fishing).**

| Location | Line | Net | Spear |
|---|---|---|---|
| Hā'ena, Kaua'i | 0.09 | 0.43 | 0.56 |
| Hanalei, Kaua'i | 0.07 | 0.96 | 0.87 |
| Kahekili, Maui | 0.09 | 0.03 | 0.30 |
| Kailua, O'ahu | – | – | – |
| Kalaupapa, Moloka'i | – | – | – |
| Kaloko-Honokōhau, Hawai'i | 0.01 | 0.07 | 0.67 |
| Kāne'ohe Bay, O'ahu | 0.27 | 0.87 | 0.93 |
| Ka'ūpūlehu, Hawai'i | 0.23 | 0.39 | 0.51 |
| Kīholo, Hawai'i | 0.62 | 1.81 | 1.79 |
| Maunalua Bay, O'ahu | 0.10 | 0.11 | 0.23 |
| Pearl Harbor, O'ahu | 0.06 | – | 0.42 |
| Puakō, Hawai'i (1980–1981) | 0.28 | – | 0.48 |
| Puakō, Hawai'i (2008–2009) | 0.15 | 1.27 | 0.23 |
| Pūpūkea, O'ahu | – | – | – |
| Waikīkī, O'ahu | 0.04 | – | 1.13 |
| Wailuku, Maui | 0.12 | 0.14 | 0.22 |
| Waimānalo, O'ahu | – | – | – |

Notes:
Some values were not available ("–"). Details for derivations of these values are provided in Supplemental Information S1.

(Fig. 3). A large proportion of total catch was reported to be composed of small coastal pelagic species (primarily *Selar crumenophthalmus*) or octopus (*Octopus cyanea* and *Callistoctopus ornatus*) at a few sites (Table 3). Octopus accounted for 21.3% of the catch in Kāne'ohe Bay (surveyed 1991–1992), and 36.6% at Kahekili (surveyed 1/2011–12/2011) (Table 3). Hanalei Bay (surveyed 1992–1993) on Kaua'i, which had the second largest annual catch, almost 40% of the catch biomass consisted of a small coastal pelagic species, *S. crumenophthalmus* that were almost all caught in nets. At Puakō, annual fisheries harvest decreased from 1980–1981 to 2008–2009 (Fig. 3). The size of the more recent creel survey at Puakō is 59% of the area of the older survey area, but the catch of the new survey was estimated to be only 29% of the total annual catch of the previous survey. In the 1980–1981 survey of Puakō, the three species with the highest harvested biomass were acute-jawed mullet (*Neomyxus leuciscus*), convict tangs (*Acanthurus triostegus*), and blackspot sergeant (*Abudefduf sordidus*). In the 2008–2009 survey at this site, the top three species harvested were convict tangs (*A. triostegus*), spotted surgeonfish (*Ctenochaetus strigosus*), and rudderfishes (*Kyphosus* spp.).

Catch-per-unit-effort estimates were generally lower in more urban and/or touristic locations, as expected (e.g., O'ahu, and Kahekili [surveyed 1/2011–12/2011] and Wailuku [surveyed 2013–2014] on Maui). Conversely, these estimates were generally higher at less densely population places, such as Hā'ena (surveyed 2009–2010) and Hanalei (surveyed 1992–1993) on Kaua'i, and Kīholo (surveyed 2012–2013) on the island of Hawai'i, with the exception of high spear CPUE at Waikīkī (surveyed 1998–2001)

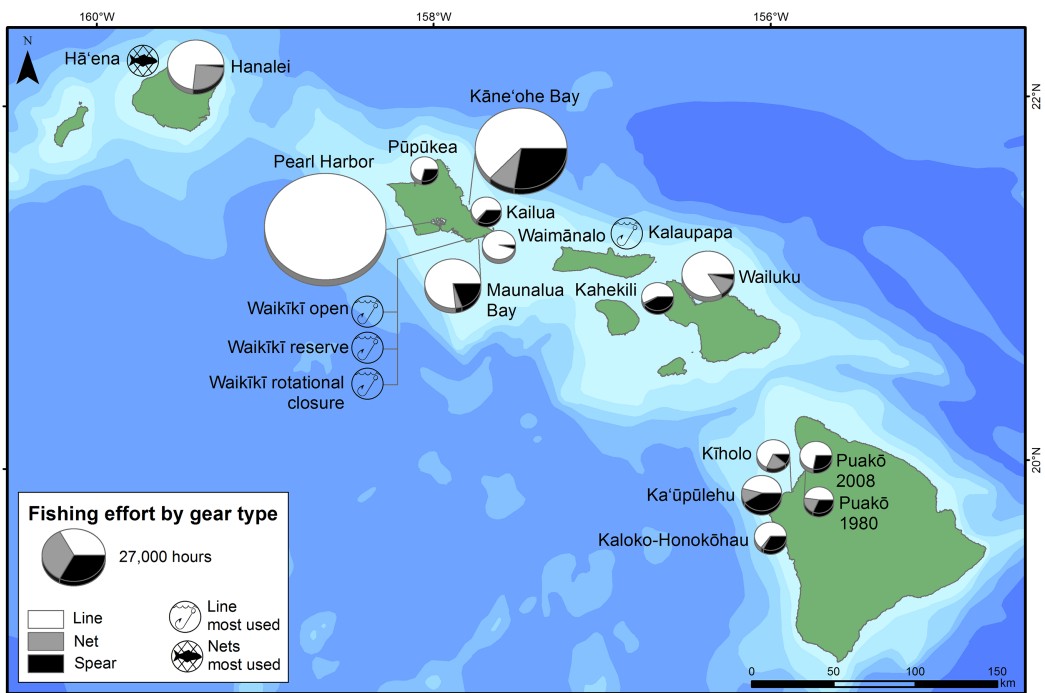

**Figure 2 Shore-based fishing effort by gear type.** Pie sizes are scaled to represent annualized estimates of total fishing effort by shore-based line, net, and spear fishing activities at each site. If annualized estimates of effort hours were not quantified for the gear types but the survey reported the most commonly used gear type (e.g., gear with highest frequency of occurrence or density of fishing activities by gear type), a symbol indicating the most commonly used gear was added to the map to document this gear preference.

(Fig. 4; Table 4). CPUE estimates were relatively high at Kāneʻohe Bay (surveyed 1991–1992), which could be due to the earlier time period and because line fishing CPUE estimates were combined for shore and boat-based activities. At Puakō, CPUE estimates decreased from 1980–1981 to 2008–2009 for both line and spear fishing by 46.4% and 52.1%, respectively (Table 3). Line fishing was almost always the least effective gear type (Fig. 4). Illegal fishing was reported at least nine of the survey locations. Violations at sites ranged from harvested undersized fish, use of illegal gears, take of prohibited species, and fishing in off-limit areas (Table 5).

## Patterns in fish flows

The majority of the fish caught was either kept or given away for home consumption (Fig. 5). Negligible proportions of the catch were reported as sold (Fig. 5). At Wailuku (surveyed 2013–2014), Maui, >40% of the catch was released, reportedly due to fish being undersized. While all the fish flow surveys quantified the proportion of fish and invertebrate biomass kept, not all surveys (e.g., Wailuku) quantified the proportion of catch that was sold. Additionally, some of the fish flow categories varied among survey locations. For example, the survey of Wailuku (surveyed 2013–2014) only reported the categories of catch that were kept and released and for surveys conducted at Hāʻena (surveyed 2009–2010) and Puakō (surveyed 2008–2009), catch used as bait was quantified separately.

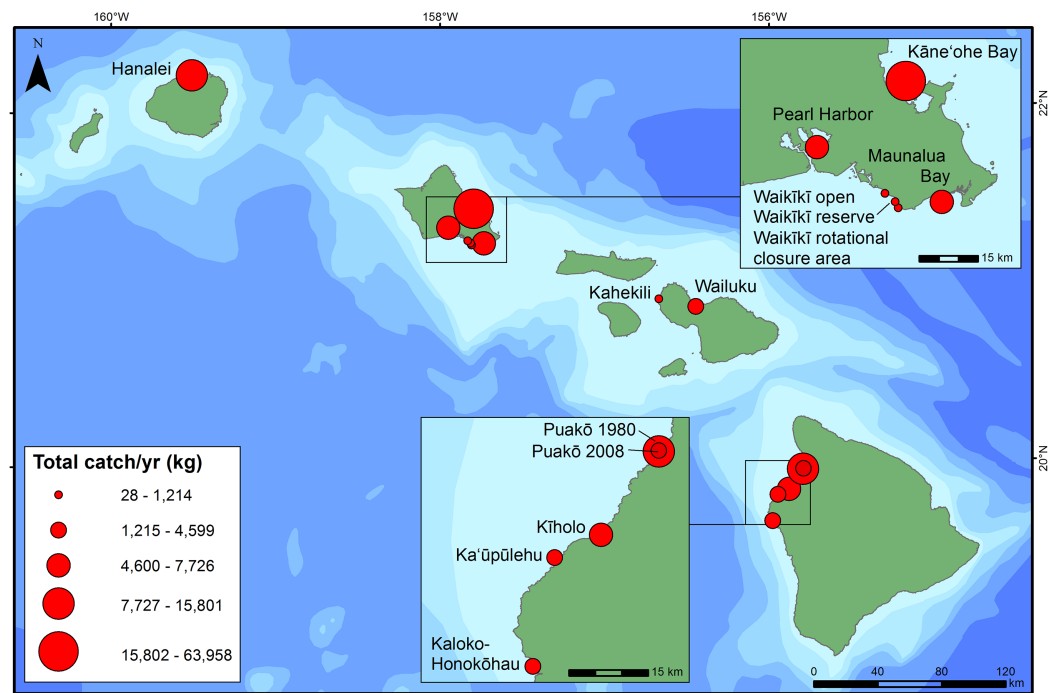

**Figure 3 Total catch per year (kg) at each site.** Circles scaled to represent total annual fisheries and invertebrate harvest at that site.

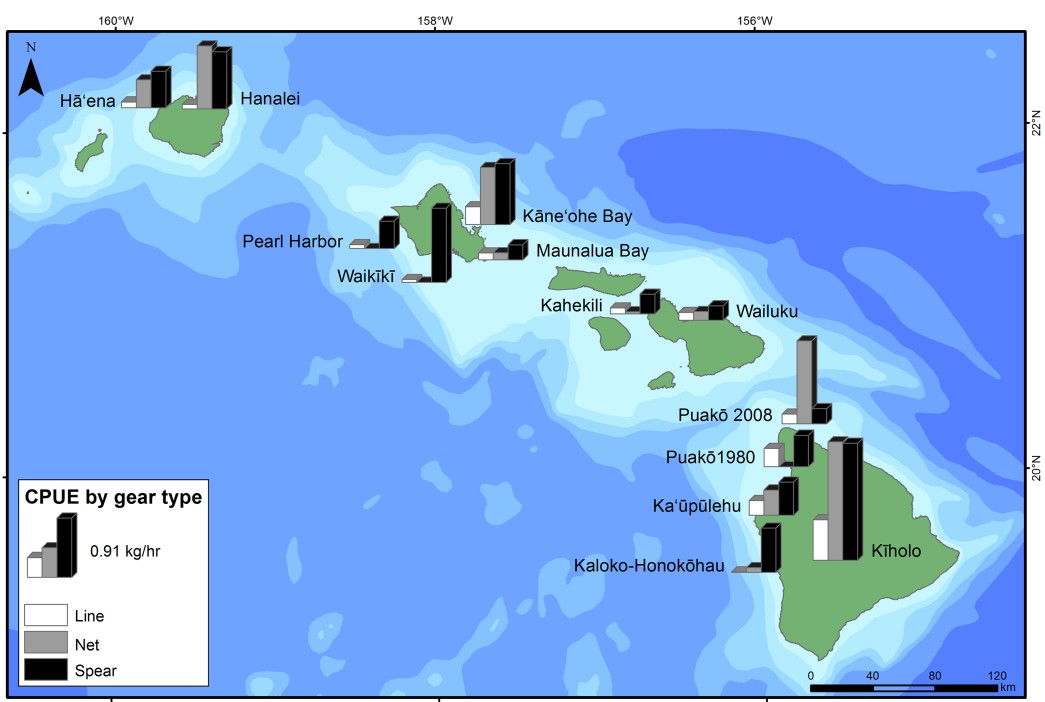

**Figure 4 Catch-per-unit-effort (CPUE—kg h⁻¹) for the three dominant shore-based fishing gears (line, net, and spear) by survey location.**

**Table 5 Location and examples of the reported illegal fishing activity reported at survey sites.**

| Location and survey period | Type of activity |
| --- | --- |
| Hanalei, Kaua'i (surveyed 1991–1992) | More than 70% of all the juvenile jacks (Carangidae) caught were below the minimum legal size |
| Kahekili, Maui (surveyed 1/2011–12/2011) | At the Kahekili herbivore management area there was illegal take of herbivorous fishes |
| Kailua, O'ahu (surveyed 2008–2013) | Illegal gill net activities were detected in 2008 and 2012 |
| Pearl Harbor, O'ahu (surveyed 2015–2016) | Spearfishing and net fishing were documented in areas where these gear types were not allowed, as well as the catch of undersized species, primarily small jacks |
| Puakō, Hawai'i (surveyed 2008–2009) | Many of the convict tangs (*Acanthurus triostegus*), parrotfishes (Scaridae) and jacks (Carangidae) that were retained were smaller than the minimum legal size |
| Pūpūkea, O'ahu (surveyed 6/2011–9/2011) | An average of 27 fishers per week illegally fish in the Pūpūkea–Waimea marine reserve |
| Waikīkī reserve and boundary areas of the reserve (surveyed 1998–2001) | Dozens of illegal spear, and pole and line fishing events were observed in the Waikīkī reserve |
| Wailuku, Maui (surveyed 2013–2014) | 33% of the fishing activity recorded was illegal and included exceeding daily allowance for marine life and using nets that were illegal size or permitted type |

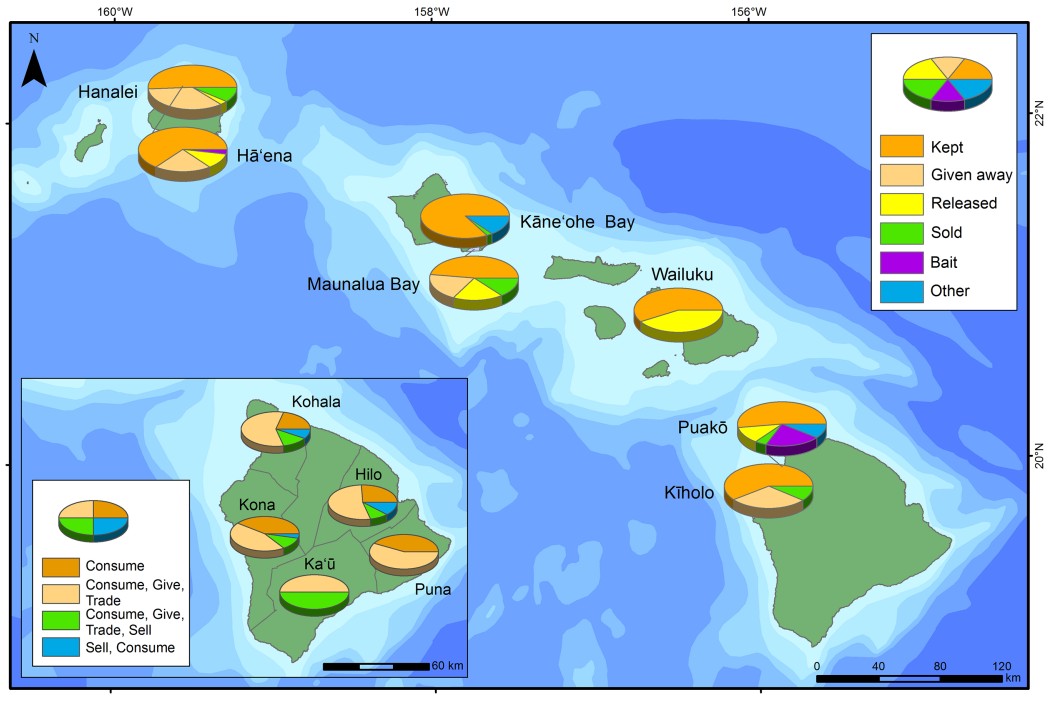

**Figure 5 Fish flow for each survey location.** Disposition of catch: kept, given away, used as bait, other, released and/or sold. In the lower left inset, data from *Hardt (2011)* on fish flows were included.

# DISCUSSION

Our first-of-its-kind regional analysis of spatial trends in nearshore fisheries based on creel surveys reveals important insights into the characteristics of coral reef fisheries in unprecedented detail. The compiled surveys comprised >10,000 h of monitoring across a diverse set of locations, with participation from local communities, state agencies, academics, and non-profit organizations. Below, we discuss the implications of our

findings for managing these complex coral reef fisheries, focusing on options available to managers to support increased ecosystem production and resilience.

## Management options for rebuilding coral reef fisheries

There are several ways to assess the sustainability of reef fisheries. While a range of yield estimates for sustainable harvest have been proposed for coral reefs (*Stevenson & Marshall, 1974*; *Dalzell, 1996*; *Newton et al., 2007*; *McClanahan et al., 2011*), by most measures the nearshore fisheries in Hawai'i are in poor health (*Friedlander & DeMartini, 2002*; *Friedlander et al., 2008*; *Williams et al., 2008*; *Nadon et al., 2015*; *Friedlander et al., 2017*). By pairing creel surveys with fish flow surveys, it is possible to assess human use patterns and drivers of behavior in nearshore fisheries in greater detail, as well as the extent of fisheries benefits to communities. Such information can help communities develop a more informed understanding of the drivers of marine resource harvest and the state of the resources. This, in turn, helps inform effective, sustainable community-based fisheries management.

Managers have the difficult task of addressing multiple drivers of fishery decline in Hawai'i, in order to sustain the important sociocultural and economic values these activities provide for local communities and visitors (*Cesar & Van Beukering, 2004*; *OmniTrak, 2011*). In order to manage these resource systems, fisheries managers have at their disposal a range of tools that can be tailored to the specific context and challenges of a fishery to sustain, and in some cases, rebuild fisheries (*Walters & Martell, 2004*; *Worm et al., 2009*; *Hilborn & Ovando, 2014*). Many of these management tools were developed hundreds of years ago by indigenous peoples, and these strategies were particularly well developed in the Pacific Islands due to their heavy reliance on, and scarcity of these resources (*Johannes, 1978*). In general, these tools include input controls (which restrict effort) and output controls (which restrict catch), which are further supported by a range of technical measures (such as monitoring, assessment, and enforcement) (*Walters & Martell, 2004*).

## Output controls

Output controls include annual catch limits, catch size restrictions, bag limits, and other limitations on catch. Allowable catch limits have been established for coral reef fisheries in Hawai'i under federal law (*WPRFMC, 2009*). However, uncertainty in catch and effort data, weak to no controls on recreational and subsistence catch, limited biological information, and conflicting management authorities have made these measures mostly ineffective. Bag limits (essentially fixed individual quotas) have been proposed for a number of prized target species in Hawai'i and these regulations have utility in making the fishing public aware of resource limitations and the large impact from the recreational and subsistence take of these species. Although they allow the resource to be shared by many, enforcement is a major problem and these restrictions may not adequately prevent overfishing owing to the lack of accurate catch and production estimates. Their application is best suited for select species with high cultural, food, and economic value along with those fisheries that target juveniles.

Minimum size limits are a basic tenant of fisheries management, allowing fish to reach reproductive size before harvest. However, for most coral reef fisheries, setting minimum size limits is constrained by the general lack of information on size at first reproduction for many, if not most, of the species harvested. Size limits are also not effective with non-selective gear such as gillnets, and enforcement of size limits is difficult since fishes are often not sold at a central market where sizes can be regulated easily. Size limits have proven to be ineffective in centralized management of coral reef fishes due to the large number of species in the fisheries, enforcement difficulties, and public awareness challenges.

## Input controls

Advances in fishing technology (GPS, monofilament nets, scuba, etc.) have greatly increased the efficiency of harvesting coral reef fishes in Hawai'i. Despite their effectiveness in Hawai'i, nets and spears are used far less often than line, which means further regulation on nets and spears could possibly reduce overall catch while not affecting the majority of fishers. Gill nets are regulated in some areas in Hawai'i (e.g., Kailua, a large portion of the south shore of O'ahu, number of locations in West Hawai'i, and the entire island of Maui) (*Division of Aquatic Resources, 2017*). Justification for these bans was the indiscriminate catch, including juvenile fishes, and a high bycatch of threatened and endangered species (e.g., sea turtles, marine mammals) (*Smith, 1993*; *Blaber et al., 2000*; *Donovan et al., 2016*). Certain methods of spear fishing such as nighttime and/or on scuba are highly efficient, particularly for parrotfishes, which sleep on the reef at night and are easily harvested at that time (*Richmond et al., 2002*; *Sabetian & Foale, 2006*). Scuba-based spear fishing is now banned in West Hawai'i, along with many Pacific Island nations and territories (*Gillett & Moy, 2006*; *Lindfield, McIlwain & Harvey, 2014*; *Division of Aquatic Resources, 2017*). Despite the selective nature of spearfishing gear, it is used rather non-selectively in many cases (*Fenner, 2012*). Restriction on or banning of gill nets, nighttime and/or scuba-based spear fishing could potentially be quite effective and should be considered for other locations as well (*McClanahan, Maina & Davies, 2005*; *McClanahan & Cinner, 2008*; *Cinner et al., 2009b*). Fishers are usually more supportive of gear restrictions than fisheries closures since they can often switch to another gear type (*McClanahan, Maina & Davies, 2005*). In addition, gear restrictions are also preferred by fishers because they are easier to circumvent than other fisheries management strategies (*Cinner et al., 2009b*).

Parrotfishes and other herbivores have been protected in a number of locations owing to their importance in reducing macroalgal abundance and enhancing the dominance of crustose corallines, which are necessary conditions for the maintenance of healthy reef communities. Management of herbivores (e.g., parrotfishes) has been successfully implemented on Maui with bag limit on parrotfishes and Kahekili's herbivore management area (*Friedlander et al., 2012*; *Williams et al., 2016*; *Division of Aquatic Resources, 2017*). These are promising solutions that could be deployed in other geographies or at a scale to further bolster reefs against impacts of climate change. Hawai'i Division of Aquatic Resources has developed and implemented numerous output

controls including size, season, and bag limit rules. However, these controls should be adapted as new knowledge emerges of the geographical variability in spawning cycles and growth characteristics of various reef fish among locations (*Schemmel & Friedlander, 2017*).

## Technical measures

Marine protected areas (MPAs), have been proven to be highly successful in conserving biodiversity globally (*Lubchenco et al., 2003*; *Lester et al., 2009*), and particularly in Hawai'i (*Friedlander et al., 2003*; *Friedlander, Brown & Monaco, 2007*). MPAs can also benefit adjacent fisheries through two primary mechanisms: increased production and export of pelagic eggs and larvae (larval spillover), and net emigration of adults and juveniles (adult spillover) (*McClanahan & Mangi, 2000*; *Gaines et al., 2010*). Within the MHI, there are numerous state-managed areas that limit fishing activities in nearshore marine waters. Existing MPAs in Hawai'i that are fully protected from fishing have higher fish biomass, larger overall fish size, and higher biodiversity than adjacent areas of similar habitat quality (*Friedlander, Brown & Monaco, 2007*; *Friedlander et al., 2014b*). These protected areas can also benefit local fisheries, as in the case of the Pūpūkea-Waimea MPA on the north shore of O'ahu, which has resulted in significant benefit for fishers through adult spillover (*Stamoulis & Friedlander, 2013*). There is much resistance to the establishment of MPAs from the fishing sector for a variety of reasons including: loss of fishing areas, displacement or marginalization of subsistence fishers, perceived loss of income and cultural access, and the long lag time before benefits are realized (*McClanahan, Maina & Davies, 2005*; *Cinner et al., 2009a*). Although not a panacea for coral reef fisheries management, MPAs in conjunction with other input and output controls are critical to sustaining fisheries and maintaining ecosystem health.

Pacific islanders traditionally used a variety of closures that were often imposed to ensure large catches for special events, or as a cache for when resources on the usual fishing grounds ran low (*Johannes, 1978*). Traditional periodic closures can be effective for short-lived taxa that reproduce quickly, but evidence across the Pacific, including Hawai'i, shows that taxa that are long-lived and reproduce later in life do not benefit from rotational closures (*Williams et al., 2006*). Rotational closures have been less successful in contemporary Hawai'i where there are few to no controls on effort once the area is open to fishing.

## Customary management

Local fisheries management that is driven and informed by traditional knowledge has been shown to be effective in certain locations in Hawai'i, as well as other locations in the Pacific (*Poepoe, Bartram & Friedlander, 2005*; *Friedlander, Shackeroff & Kittinger, 2013*; *Severance et al., 2013*; *Levine & Richmond, 2014*; *Birkeland, 2017*). A diverse range of management options needs to be developed through a collaborative approach. There is a strong movement in Hawai'i toward decentralized fisheries management, with a revitalization of community-based fishery management based on customary practices and knowledge (*Friedlander, Shackeroff & Kittinger, 2013*; *Vaughan & Vitousek, 2013*),

including a recent legal mandate for collaborative management between the state and local communities to establish community-based subsistence fishing areas (*Kittinger et al., 2012*; *Ayers & Kittinger, 2014*). There are over 20 community initiatives currently active in Hawai'i (*Ayers & Kittinger, 2014*), which is among the most promising developments in nearshore fishery management. In Hawai'i, the cultural diversity and isolation of the islands lead to many expressions of self-determination; one of those expressions is the desire for "local production for local consumption, under local control" (*Loke & Leung, 2013*). At all sites where fish flow surveys were conducted, the majority of fish caught was either kept or given away for local consumption, demonstrating the high food security and cultural value of these non-commercial subsistence/recreational fisheries for the people of Hawai'i, particularly in rural areas.

Direct translation of traditional practices into a modern management context is often not possible for political and historical reasons. Current management strategies are often an adaptation and melding of traditional with the contemporary (*Cinner & Aswani, 2007*; *Shackeroff & Campbell, 2007*; *Jokiel et al., 2011*; *Ayers & Kittinger, 2014*). Movement towards the establishment of more co-management arrangements is also driven by recent findings that locations under community-based management have similar amounts or greater fish biomass compared to no-take protected areas (*Friedlander, Shackeroff & Kittinger, 2013*). Both of these management regimes harbor higher biomass than partially protected or completely open-access areas, clearly indicating that community-managed areas can be effective in providing positive ecological outcomes by sustaining both ecosystems and ecosystem benefits (*Friedlander, Shackeroff & Kittinger, 2013*).

## Local monitoring as a tool for reducing illegal fishing

Illegal, unregulated, and undocumented (IUU) fishing activities are a major cause of negative fishing impacts in coral reefs and other marine environments (*Sumaila, Alder & Keith, 2006*), with a lack of awareness on local regulations, as well as weak law enforcement, acting as key contributing factors for lack of compliance (*Bergseth, Russ & Cinner, 2015*). In Hawai'i, survey results show that fishers identify weak enforcement of fisheries laws as a top threat to fishery resources (*OmniTrak, 2011*), and lack of compliance and weak enforcement is one of the priority threats to Hawaiian coral reefs and a key capacity gap.

Our results highlight numerous illegal fishing activities occurring across the MHI (Table 5). Illegal fishing activities were reported at at least half of the locations surveyed, with frequent violations at some locations. Violations at sites ranged from harvested undersized fish, take of prohibited species, use of illegal gears, and fishing in off-limit areas. Instances of illegal fishing and the spatial and temporal patterns of fishing catch and effort have important management implications, and such trends help guide strategies to optimally monitor fisheries given logistical limitations (e.g., limited time, equipment, and personnel to monitor vast amount of area). For this reason, managers and scientists cannot monitor the entire geographic areas of most coral reef fisheries as intensely as needed. In these situations, local monitoring efforts are critical to inform place-based

management (*McClanahan & Mangi, 2004*; *McClanahan et al., 2006*). However, these local monitoring efforts also need to be aggregated into broader analyses of temporal and spatial scales for managers to gain insights about fishery trends and appropriate management approaches.

Community monitoring via creel surveys can help detect and quantify the extent of illegal activities (e.g., Kahekili (*Friedlander et al., 2012*)), but also potentially reduce them (Kīholo (*Kittinger et al., 2015*)). More enforcement capacity will be required in order to better support existing input and output regulations as well as new rules being advanced through community-led initiatives and state-wide initiatives (such as scoping the feasibility of a licensing system). Creel surveys could be better integrated with these educational and enforcement programs to inform these efforts.

To best monitor legal and IUU fishing, we need to determine the most effective survey approach for a given set of personnel, geographies, and available resources. For example, many coastal areas in Hawai'i are expansive and relatively undeveloped. Access points to these areas are generally indistinct and parking haphazardly on the side of a road is common. Local fishers often prefer to utilize these areas where resources are generally not as depleted. In addition to many line fishers, spear and net fishers tend to favor remote areas where capture success is likely greater due to higher resource availability. Based upon the general characteristics of the coastal areas and the diversity of shore-based fisheries in Hawai'i, the roving survey is often considered the more suitable survey method to collect shoreline fishing information. Also, given that fishing effort is generally higher on weekends and holidays during non-winter months, optimally allocating limited effort for monitoring to those times and locations with higher fishing pressure, could possibly lead to better coverage of fishing activities as well as better enforcement of current regulations. Community monitoring can detect illegal activities, but also potentially reduce them (*Kittinger et al., 2015*). Also, more enforcement capacity will be required to better support existing input and output regulations. Some promising initiatives include the community fisheries enforcement unit, Makai Watch, and other existing initiatives in Hawai'i. Creel surveys could be better integrated with these educational and enforcement programs.

## Improving nearshore fisheries monitoring and evaluation

On many coral reefs across the Pacific, including Hawai'i, there is limited capacity for fisheries monitoring (*Pauly & Zeller, 2014*), thus making fishing effort and total annual catch poorly understood and difficult to quantify (*Zeller et al., 2006*; *McCoy, 2015*; *Gillett, 2016*). These realities behoove us to identify cost-effective and accurate monitoring tools and survey instruments to appropriately track ecological and social aspects of small-scale tropical fisheries, the results of which can successfully inform adaptive state and community-level fisheries management. To address this, Hawai'i is currently exploring a range of options, including requiring recreational fishing licenses and/or reporting of recreational/non-commercial fisheries, which would provide critical information on non-commercial catch and effort (*Noncommercial Fisheries Licensing Steering Committee, 2016*).

In many of the locations, fishing effort was typically higher on the weekends and holidays than on weekdays (*Hayes et al., 1982*; *Friedlander & Parrish, 1997*; *Koike et al., 2015*). Along north facing shores, fishing effort was constrained in winter due to large surf (*Everson & Friedlander, 2004*; H. Koike, J. Carpio, A.M. Friedlander, 2014, unpublished data). Many locations in Hawai'i, particularly sandy shores and embayments, experience higher fishing effort during summer (June–August) when juvenile goatfishes aggregate in mass very close to shore (*Kamikawa, 2016*). Summer months also experienced higher fishing pressure because school is not in session and weather conditions are typically more favorable, allowing more people to spend more time fishing with their families.

We recommend that creel surveys be conducted in a more standardized fashion, more regularly, and at additional sites, to provide more standardized data that has better spatial and temporal coverage, which will allow for more robust analyses. This is critical to address a variety of management issues. First, fishing effort and catch can change, not only across space, but also over time. For example, estimates were quite different at Puakō between surveys of the area in 1980–1981 and 2008–2009. Yet the exact location and size of the study site varied between surveys, which makes comparison more challenging. Nevertheless, current estimates of catch and effort could also be quite different for Hanalei Bay and Kāne'ohe Bay, which were surveyed in 1992–1993 and 1991–1992, respectively. Therefore, repeat sampling of the same areas through time is needed. The surveys of 18 sites were conducted over decades, which does not allow for spatial and temporal aspects to be disentangled, which would have great importance for conservation and management actions. Furthermore, the information reported from these surveys hampered full analysis of the data. For example, some creel surveys only reported some of the species and/or family-level catch data. Another example is that some studies reported effort in semi-quantitative measures (e.g., frequency of occurrence) or number of fishers rather than the more preferred form of gear-hours, which better allows for estimates of effort and comparison among sites. These issues of standardization hindered the ability to conduct some analyses and forced us to look at coarse measures such as overall catch and effort. The execution of future creel surveys in a more standardized fashion will be of greater benefit to research and management. We recommend that when these data are available, analyses are conducted to the species-level, comparisons of catch and effort on a finer scale (e.g., within a survey area, better differentiation of gear types [e.g., cast net, laynet, surround nets]). Additionally, we recommend examining how fishing effort and catch vary across space and time, which was not possible in our study due to limited and non-standardized reporting of the creel surveys that were previously conducted.

## CONCLUSION

Reef fisheries across the planet provide important benefits to communities that are threatened by a range of factors including overfishing, pollution, invasive species, climate change, and other threats. Our research reveals regional patterns in harvesting in these complex fisheries, which can be used to inform a range of regulatory approaches to rebuild these fisheries. Together with reductions in land-based pollution, invasive

species control, and other measures, managing these local drivers can help reefs be more resilient to climate change, a global threat that is undermining reef ecosystems across the planet (*Hughes et al., 2017*). Site-based survey methods that are led by community groups can also provide benefits in co-learning, reduction of illegal and harmful activities, and building community capacity, which are necessary for effective stewardship (*Kittinger, 2013*). In an era where the threats to reefs and their associated fisheries are escalating due to overfishing, pollution, and climate change (*Bell, Johnson & Hobday, 2011*), local efforts must be embedded into broader regional management efforts if reefs and the benefits they provide to people are to survive.

## ACKNOWLEDGEMENTS

We would like to thank all the communities, organizations, funders, and individuals that were involved with any stage of the surveys, the editor, and two anonymous reviewers. We thank Kaylyn McCoy for providing data from her Master's thesis research at the University of Hawai'i and we thank Joey Lecky and Hla Htun for GIS data that we built upon.

### Funding

This work was supported by the National Oceanic and Atmospheric Administration's Saltonstall-Kennedy program, through award NA15NMF4270332, granted to Conservation International. The funders had no role in study design, data collection and analysis, decision to publish, or preparation of the manuscript.

### Grant Disclosures

The following grant information was disclosed by the authors:
National Oceanic and Atmospheric Administration's Saltonstall-Kennedy Program: NA15NMF4270332.
Conservation International.

### Competing Interests

Dr. David Delaney is the owner of Delaney Aquatic Consulting LLC.

### Author Contributions

- David G. Delaney conceived and designed the experiments, performed the experiments, analyzed the data, wrote the paper, prepared figures and/or tables, reviewed drafts of the paper.
- Lida T. Teneva conceived and designed the experiments, performed the experiments, analyzed the data, wrote the paper, reviewed drafts of the paper.
- Kostantinos A. Stamoulis conceived and designed the experiments, performed the experiments, analyzed the data, wrote the paper, prepared figures and/or tables, reviewed drafts of the paper.

- Jonatha L. Giddens conceived and designed the experiments, performed the experiments, analyzed the data, wrote the paper, reviewed drafts of the paper.
- Haruko Koike conceived and designed the experiments, performed the experiments, analyzed the data, reviewed drafts of the paper.
- Tom Ogawa conceived and designed the experiments, performed the experiments, analyzed the data, wrote the paper, reviewed drafts of the paper.
- Alan M. Friedlander conceived and designed the experiments, performed the experiments, analyzed the data, wrote the paper, reviewed drafts of the paper.
- John N. Kittinger conceived and designed the experiments, performed the experiments, analyzed the data, wrote the paper, reviewed drafts of the paper.

## Data Availability

The research in this article did not generate any new data or code. All data were published before or were generated from existing data as described in Supplemental Information (S1).

## Supplemental Information

Supplemental information for this article can be found online at http://dx.doi.org/10.7717/peerj.4089#supplemental-information.

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
