# Peer review of "Patterns in artisanal coral reef fisheries revealed through local monitoring efforts"

_PeerJ, doi:10.7717/peerj.4089_

## Round 0.1 · original submission · Major Revisions

The reviewers and I are very supportive of this paper that makes a valuable contribution. However, due to the nature of the data there a number of caveats that need to be made more explicitly and discussed. I would urge the authors to go through reviewers' comments and consider the corrections they suggest.

Reviewer 1 ·

Basic reporting

The authors did a great job in reporting on the disparate data of creel surveys. The MS is clearly written although could benefit from an additional read-through to streamline the text as some is duplicated. Sufficient background is included to provide context and understand the usefulness of non-commercial data to sustainably manage near-shore fisheries.

It appears that the reported data is not the raw creel survey data but the estimated data by each of the literature sources. The results could benefit from adding SEs especially the CPUE data to improve the spatial comparisons. Of course this would mean that the raw data is accessible from the 18 literature resources used in this review.

Experimental design

The title indicates guidelines of ‘best practices’ but the MS seems more a compilation of the creel survey data from the various literature sources and misses in-depth analyses. ‘Best practices’ seem to be, reduced effort on those gear types that are most effective overall (highest CPUE) while low in total gear-hours. However, as the introduction and discussion mention, knowing the human use (e.g. a recreational fisherman fishing with hook and line for fun versus a fisherman putting out his laynet to catch a living) is of course more important than establishing the general overall gear-hours per gear type. Furthermore, gear types do not always target the same species e,g, catching parrotfish with spearfishing versus catching octopus with the same gear type has very different ecological consequences. The MS would benefit from an evaluation of the species caught and how a reduction on those species might benefit the ecosystem.

It is a great and laudable effort to compile all these data in one place. However, these data could benefit from standardization to make use of them more efficient. For example, In reporting effort, gill nets and surround nets were not included while total catch does include the catches of these methods (e.g. for Kane’ohe 58% of the total catch was from surround/gill nets and in Hanalei these gear types comprised 69% of the total catch). The MS would further benefit from separating out the invertebrates (e.g. the catches in Kane’ohe were largely comprised of invertebrates (octopus) caught by spear fishing – hence high spear fishing effort for that location), akelu/opelu (e.g. Hanalei catches comprised of 69% akule/opelu) and coral reef fish. This would benefit managers trying to regulate the fishery based on gear types/ species / species complexes as regulations for octopus could be very different from e.g. opelu. Moreover, it would facilitate comparing just reef fish (or invertebrates or opelu/akule) spatial patterns which I would argue is more interesting to reef managers and the fishing community then the overall total catch or effort where this information is lost.

The data set comprises a temporal time span from 1980 – 2015 with most data sets from the latest 10 year. The Puako data showed that there can be very large temporal changes in the fishery over a 30 year time span as the authors mention as well in the discussion. The authors acknowledge that over a time span of 20 or 10 years similar changes could have occurred at other reefs. Maybe results can be binned by survey year to see if temporal patterns emerge which might be more important than the spatial patterns? Even if no additional analyses are done, it would be better to include the year of the survey in the results for clarity.

Validity of the findings

In general, some of the conclusions are not well backed up by arguments/findings. For example, the statement that "The findings can directly improve the design of more cost-effective and standardized creel survey approaches that can be instituted to facilitate more informed fisheries management in Hawai‘i and beyond" is not well argumented. It would be very interesting if the authors include the "How". Knowing that fishing effort, catch and CPUE estimates varies across space could be partly due to the differences in target species, the inclusion of gill net & surround net in some places, or the temporal differences in survey years, it is unclear to me how these findings can improve a more cost-effective creel survey.
As I think the authors are aware of, there is a movement to make reporting of recreational/non-commercial fisheries obligatory (through license fees) which of course would also give us information on the non-commercial catches and efforts once implemented. The MS would improve from including these efforts and maybe comparing the pros and cons of both. Additionally, NMFS-PIFSC has recently published a technical memorandum with stock assessments of 27 reef fish species. These could help inform the setting of ACLs. It is unclear and not elaborated on why the authors think that ACLs are 'most effective' for high valued invertebrates.

Lastly, I am surprised that the authors find the results from the fish flow (i.e. that most of the catch is not sold) 'surprising' as this has been reported in pretty much all literature on non-commercial fishing.

Additional comments

Some more specific comments:
Ln 42 “Surprisingly’? In almost all articles related to non-commercial fishing it is reported that a great majority of the catch is not sold e.g. Friedlander and Parrish 1997, Friedlander et al 2012. Suggest to re-word.
Ln 81-83: This sentence is of course true but some head-way has been made with the stock assessment of 27 reef fishes and is worth mentioning. (Nadon et al 2017 NMFS-PIFSC Tech Memo #60)
Ln 96-97. This sentence seems to imply that all creel surveys are designed by community members which is not the case for most creel surveys mentioned in the tables. Suggest to delete last part of the sentence (everything after the second comma)
Ln 117-119: is it not the lack of obligatory reporting of non-commercial fishing that hinders the data collection more so than that this fishery is non-commercial? In other words, the fisherman do not have to report their catches, (unlike the commercial fisherman) so don’t do it unless they participate in the MRIP data collection by DAR, and therefore data is limited.
Ln 197 and ln 198 seem to say the same?
Ln 197, 198. From table 3 it seems that line fishing is 79% (not 94%), spear 14% (not 0%) and net 7%? Please clarify discrepancy.
Ln 208 Table 4. Without having checked all refs, it seems that for Kaneohe, the CPUE is reported for shore and boat based line fishing whereas the methods say that boat-based is not included. Not sure about others but suggest to review your methods/results so they are in line with the original literature. I understand the info given in the creel survey papers might be not sufficient to separate them out but if that is the case this should be mentioned so we know the table does not compare like with like.
Ln 227. Figure 3. These catches include gill net and surround net for some locations (e.g. Kaneohe Bay and Hanalei) which makes them not comparable with other locations that do not include the catches of those gear types. Suggest to adjust catches to make them comparable.
Ln 259 “Surprisingly” see comment earlier
Ln 282-283. Some of this variation is because of inclusion/exclusion of gear types, targeted species (e.g. opelu v octopus) and would be more interesting if the authors can specify their results separately for reef fish, invertebrates, and akule/opelu.
Ln 283-285. Can the authors expand on how the findings can improve cost-effect survey design?
Ln 287. Can the authors explain how the define “health”.
Ln 302. Isn’t a spatial measure also an input control as it restricts effort?
Ln 307. What is meant by “inputs” (other than fishing technology and effective gear types)?
Ln 308. What is meant by “outputs”, catches?
Ln 343. Why are ACLs ‘most effective’ for a high value invertebrates? Pls expand.
Ln 383-385. Suggest to include Jokiel et al 2011 Journal of Marine Biology where the authors compare traditional with western management.
Ln 399. There is a movement to make reporting of recreational/non-commercial fisheries obligatory which of course would also give us information on the non-commercial catches and efforts once implemented. Might be worth mentioning.

Reviewer 2 ·

Basic reporting

L1-2: The title seems to imply some kind of comparison between different monitoring/monitoring approaches. The paper focused mostly on characterizing fishery and perhaps this should be reflected in title instead?

L34: This “collaborative” dimension isn't very clear from methods/results. How does the approach mentioned in the study promotes/facilitates collaboration? For example, aren’t fishers simply asked questions (and if so, is that a collaborative process?)?

L41: The word “rampant” seems to imply some kind of magnitude or direction has been assessed. Perhaps reword?

L42: Why “surprisingly”? Is this different from most similar fisheries? Clarify or reword.

L53: Perhaps you mean “services” instead of “values” here?

L95: How is this participatory? Clarify as it currently seems information is simply requested of them.

L124-125: I think this should be clarified in abstract as well as mentioning how this study differs from previous studies (e.g. by putting everything together, study is able to draw comparisons, identify regional patterns, etc).

L126: What do you mean by “intercept” survey? Clarify.

L147: Can't read full label in first column.

L152: how do creel surveys differ from the interview-based surveys mentioned below then? Unclear.

L162: Redundant? Interviews can only be undertaken with participants… rephrase.

L168: What's this reference meant to support here? Also, if in review, perhaps it's not available to readers anyway?

L259: See my comment above regarding term “surprisingly”.

Experimental design

L129-137: What's rationale for inclusion of each different area? How were these selected? Are there any key differences among them that might be useful for interpreting results?

L180: what about potential temporal effects? Does it make sense putting everything together? Some consideration or discussion required.

L217 and others: I think these differences among study areas need to be stated much earlier on manuscript.

Validity of the findings

L253: How was this information obtained? Clarify in methods. Also, examples don’t provide specific time periods or magnitude of problem so difficult to use them to assess situation (although every useful for illustrating potential threats). Examples should be more specific (for example, “33% of the fishing activity recorded was illegal” refers to how many fishers? When?)

Additional comments

The manuscript “Patterns in artisanal coral reef fisheries reveal best practices for monitoring and management” by Delaney et al. provides useful insights about small-scale fisheries in Hawai‘i with clear implications for natural resource management. However, several key issues should be addressed before further consideration for publication.

Firstly, the contribution of the study should be made clearer. To do so, I think the authors should: (1) explicitly mention their general approach in terms of data gathering from previously published/unpublished studies much earlier on the manuscript (e.g. in the abstract) and explain how this study differs/builds upon those (see specific comment); and (2) clarify how this study aims to contribute in terms of data collection/monitoring protocols. For example, are creel surveys widespread in fisheries studies or only occasionally used (and if so, are the authors suggesting this should be elsewhere?). Either way, this should be clearer so that readers understand how this study fits within wider literature and fisheries assessments and how it contributes to advancing any knowledge gaps.

Secondly, in the introduction, more focus should be given to the types of challenges and barriers involved in small scale fisheries monitoring and managing in data-poor contexts as well as different approaches used to address this issue. I think that would be useful for putting this study into wider context. I also felt the introduction was a bit repetitive (e.g. the livelihoods/social/economic/etc. issue was mentioned several times throughout introduction) and occasionally jumped between general and specific issues with somehow unclear flow. I recommend restructuring and rewriting some sections so that it reads better in terms of narrative and is more focused around key management/monitoring challenges.

Finally, while I commend the authors for the work involved in bringing all the survey information together, I think the potential caveats in doing so haven’t been properly mentioned and discussed. For example, it’s clear from reading appendix material that a lot of different assumptions and steps were involved in producing estimates from different data sources. I thank the authors for providing a detailed description in the appendix. Nevertheless, some discussion of potential biases due to different survey approaches, sampling period (e.g. the authors pooled together data from different decades), etc., is required.

---

## Round 0.2 · Minor Revisions

As you can see the reviewer is happy with the corrections you have made and only noted a few remaining minor points. Once these are made I'd be happy to accept.

Reviewer 1 ·

Basic reporting

Well written some minor comments:

Ln 329 Define “health” (health of reef fisheries and ‘health of associated ecological community” – maybe replace by “productivity of reef fisheries” and “status of associated ecological community” although I am not sure why this sentence is added as the rest of this section focused on fishing and nothing is mentioned about “numerous ways to assess the health”. Suggest to delete or add a sentence that includes refs for the latter part of this sentence.
Ln 331-336 This paragraph comes a bit out of the blue – the results show that the catches are hardly sold (fig 5) and there is no mention about the composition of commercial and non-commercial catches so the reader has no idea how much of the reef fish is actually sold. Maybe the authors could start this paragraph with a comparison of commercial and noncommercial fisheries? And back up these claims by references.
Ln 378, 379 The refs used to back up this sentence seem not appropriate. Donovan et al 2016 has references therein that are appropriate and I would recommend to use them. DAR 2017 has regulations but no justifications (or at least not that I could find on the linked website).
Ln 398 a scale or scales; against impacts of climate change
Editorial comments
Ln 86 “remains” should be “remain”
Ln 225 add “they” compose
Table 5, Pupukae “illegally fish”
Ln 468 at at least

Experimental design

no comment

Validity of the findings

no comment

Additional comments

The authors have done an excellent job in addressing my previous concerns and I look forward to seeing this MS being published.

---

## Round 0.3 · accepted · Accept

Many thanks for making the correction. Looking forward to seeing it published.